# Linear-Time Graph Neural Networks for Scalable Recommendations

## ABSTRACT

In the era of information explosion, recommender systems are vital tools for delivering personalized recommendations for users by forecasting their future behaviors based on historical user-item interactions. To model these interaction behaviours, Graph Neural Networks (GNNs) have remarkably boosted the prediction performance of recommender systems due to their strong expressive power of capturing high-order information in user-item interactions through multi-layer embedding propagations. Nonetheless, classic Matrix Factorization (MF) and Deep Neural Network (DNN) approaches still dominate real-world applications for large-scale recommendations due to their scalability advantages. Despite the existence of acceleration solutions, it remains an open question that whether GNN-based recommender systems can scale as efficiently as classic MF and DNN methods. In this paper, we propose a Linear-Time GNN (LTGNN) to scale up GNN-based recommender systems to achieve comparable scalability as the classic and efficient Matrix Factorization approaches while maintaining the powerful expressiveness for superior prediction accuracy. Extensive experiments and ablation studies are presented to validate and understand the effectiveness and scalability of the proposed algorithm.

## CCS CONCEPTS

• **Information systems** → **Collaborative filtering**.

## KEYWORDS

Collaborative Filtering, Recommendation, Graph Neural Networks, Scalability

## 1 INTRODUCTION

In an era of information explosion, recommender systems are playing an increasingly critical role in enriching users' experiences with various online applications, due to their remarkable abilities in providing personalized item recommendations. The main objective of recommender systems is to predict a list of candidate items that are likely to be clicked or purchased by capturing users' potential interests from their historical behaviors [24]. One of the most prevailing techniques in modern recommender systems is collaborative filtering (CF), which leverages the patterns across similar users and items to predict the users' preferences.

As one of the most representative CF methods, matrix factorization (MF) techniques are introduced to represent users and items in a low-dimensional embedding space by encoding the user-item interactions matrix. After the emergence of MF models, a remarkable stream of literature has made great efforts to improve the expressive capability of user and item representations. As discussed in many previous studies [24, 48, 49], we can divide these attempts into two branches based on their modeling ability of user-item interaction graphs. First, most early approaches in collaborative filtering focus on the *first-order connectivity* of users and items, such as item similarity models [25, 32, 35] and deep neural networks (DNNs) [26, 53]. Second, due to the intrinsic limitation of modeling high-order connectivity in early CF models, recent years have witnessed a rising interest in graph neural networks (GNNs) in recommendations. To be specific, GNN-based CF models encode both *local and long-range collaborative signals* into learning user and item representations by iteratively aggregating embeddings along local neighborhood structures in the interaction graph [14, 24, 48], showing their superior performance in modeling complex user-item interaction graphs.

Despite the promising potential of GNNs in modeling high-order information in interaction graphs, GNN-based CF models have not been widely employed in industrial-level applications majorly due to their scalability limitations [23, 56]. In fact, classic first-order models like MF and DNNs are still playing major roles in real-world applications due to their computational advantages, especially in large-scale industrial recommender systems [9, 12, 42]. In particular, the computation complexity for training these first-order models such as MF and DNNs is *linear* to the number of user-item interactions in the interaction matrix [27], while the computation complexity of training GNN-based CF models is *exponential* to the number of propagation layers or *quadratic* to the number of edges (as will be discussed in Section 2.3).

In web-scale recommender systems, the problem size can easily reach a billion scale towards the numbers of nodes and edges in the interaction graphs [30, 31, 45]. Consequently, it is essential that scalable algorithms should have nearly linear or sub-linear complexity with respect to the problem size. Otherwise, they are infeasible in practice due to the unaffordable computational cost [44]. While numerous efforts have continued to accelerate the training of GNN-based recommender systems, including two main strategies focusing on neighbor sampling [17, 38, 56] and design simplification [6, 24, 50], none of them can achieve the linear complexity for GNN-based solutions, leading to inferior efficiency in comparison with conventional CF methods such as MF and DNNs. There is still an open question in academia and industry: *Whether GNN-based recommendation models can scale linearly as the classic MF and DNN methods, while exhibiting stronger modeling expressiveness and prediction performance.*

In this paper, our primary objective revolves around *1) preserving the strong expressive capabilities inherent in GNNs* while simultaneously *2) achieving a linear computation complexity* that is comparable to traditional CF models like MF. However, it is highly non-trivial to pursue such a scalable GNN design, since the expressive power of high-order collaborative signals lies behind the number of recursive aggregations (i.e., GNN layers). Moreover, the embedding aggregation over a large number of neighbors is highly costly. To achieve a non-trivial linear computation complexity comparable to classic MF and DNN methods, we propose a novel

implicit graph modeling for recommendations with the *single-layer propagation* model design and an *efficient variance-reduced neighbor sampling* algorithm. Our contributions can be summarized as follows:

- We provide a critical complexity analysis and comparison of existing collaboration filtering approaches, and we reveal their performance and efficiency bottlenecks.
- We propose a novel GNN-based model for large-scale collaborative filtering in recommendations, namely LTGNN (Linear Time Graph Neural Networks), which only incorporates *one propagation layer* while preserving the capability of capturing long-range collaborative signals.
- To handle large-scale user-item interaction graphs, we design an efficient and improved *variance-reduced neighbor sampling* strategy for LTGNN to significantly reduce the neighbor size in embedding aggregations. The error caused by neighbor sampling is efficiently tackled by our improved variance reduction technique.
- We conduct extensive comparison experiments and ablation studies on three real-world recommendation datasets, including a large-scale dataset with millions of users and items. The experiment results demonstrate our proposed LTGNN significantly reduces the training time of GNN-based recommendation models while preserving the capacity to uphold recommendation performance on par with previous GNN models. We also perform detailed time complexity analyses and ablation studies to show our superior efficiency.

## 2 PRELIMINARIES

This section presents the notations used in this paper and briefly introduces some preliminaries about the embedding propagation in GNN-based collaborative filtering algorithms and the computation complexities of popular collaborative filtering models.

### 2.1 Notations and Definitions

In personalized recommendations, the historical user-item interactions can be naturally represented as a bipartite graph $\mathcal{G} = (\mathcal{V}, \mathcal{E})$, where the node set $\mathcal{V}$ includes $n$ user nodes $\{v_1, \cdots, v_n\}$ and $m$ item nodes $\{v_{n+1}, \cdots, v_{n+m}\}$, and the edge set $\mathcal{E} = \{e_1, \cdots, e_{|\mathcal{E}|}\}$ consists of undirected edges between user nodes and item nodes. It is clear that the number of undirected edges $|\mathcal{E}|$ equals to the number of observed user-item interactions $|\mathcal{R}^+|$ in the training data (i.e., $|\mathcal{E}| = |\mathcal{R}^+|$). The graph structure of $\mathcal{G}$ can be denoted as the adjacency matrix $A \in \mathbb{R}^{(n+m) \times (n+m)}$, and its diagonal degree matrix are denoted as $D$. The normalized adjacency matrix with self-loops is defined as $\tilde{A} = (D+I)^{-\frac{1}{2}}(A+I)(D+I)^{-\frac{1}{2}}$. We use $\mathcal{N}(v)$ to denote the set of neighboring nodes of a node $v$, including $v$ itself. In addition, the trainable embeddings of user and item nodes in graph $\mathcal{G}$ are denoted as $E = [e_1, \ldots, e_n, \; e_{n+1}, \ldots, e_{n+m}]^T \in \mathbb{R}^{(n+m) \times d}$, where its first $n$ rows are $d$-dimensional user embeddings and its $n+1$ to $n+m$ rows are $d$-dimensional item embeddings.

In the training process of GNN-based collaborative filtering models, we use $(E_l^k)_B$ or $(e_l^k)_v$ to denote an embedding matrix or a single embedding vector, where $k$ is the index of training iterations and $l$ is the index of propagation layers. The subscript $(\cdot)_B$ or $(\cdot)_v$ denotes the embedding for a batch of nodes $B$ or a single node $v$.

## 2.2 Mini-batch Training

To provide effective item recommendations from user-item interactions, a typical training objective is the pairwise loss function. We take the most widely adopted BPR [41] loss as an example:

$$\mathcal{L}_{BPR} = \sum_{(u,i,j) \in O} -\ln \sigma(\hat{y}_{u,i} - \hat{y}_{u,j}), \tag{1}$$

where $O = \{(u, i, j) | (u, i) \in \mathcal{R}^+, (u, j) \in \mathcal{R}^-\}$ denotes the pairwise training data. $\mathcal{R}^+$ and $\mathcal{R}^-$ denotes observed and unobserved user-item interactions. In practice, the training objective $O$ is hardly evaluated in a full-batch setting due to the large number of user-item interations [24, 48]. Therefore, mini-batch training is a common choice that splits the original objective $O$ into multiple components $\Omega = \{O_{(u_1,i_1)}, O_{(u_2,i_2)}, \cdots, O_{(u_{|\mathcal{R}^+|},i_{|\mathcal{R}^+|})}\}$, where $O_{(u_r,i_r)} = \{(u_r, i_r, j) | (u_r, j) \in \mathcal{R}^-\}$ contains all the training data including positive and negative samples for a specific interaction $(u_r, i_r)$. In each training iteration, we first sample $B$ interactions from $\mathcal{R}^+$, which is denoted as $\hat{\mathcal{R}}^+$, satisfying $|\hat{\mathcal{R}}^+| = B$. Afterward, we create the training data for $\hat{\mathcal{R}}^+$ by merging the corresponding components in $\Omega$, which can be denoted as $\hat{O}(\hat{\mathcal{R}}^+) = \bigcup_{(u,i) \in \hat{\mathcal{R}}^+} O_{(u,i)}$. Thus, the mini-batch training objective can be formalized as follows:

$$\hat{\mathcal{L}}_{BPR}(\hat{\mathcal{R}}^+) = \sum_{(u,i,j) \in \hat{O}(\hat{\mathcal{R}}^+)} -\ln \sigma(\hat{y}_{u,i} - \hat{y}_{u,j}). \tag{2}$$

In each training epoch, we iterate over all user-item interactions in $\mathcal{R}^+$, so the mini-batch training objective $\hat{\mathcal{L}}_{BPR}$ needs to be evaluated for $|\mathcal{R}^+|/B$ times (i.e., $|\mathcal{E}|/B$ times).

## 2.3 GNNs and MF for Recommendations

In this subsection, we will briefly introduce MF and two representative GNN-based recommendation models including LightGCN [24] and PinSAGE [56], and discuss their computation complexity.

**LightGCN**. Inspired by the graph convolution operator in GCN [34] and SGC [50], LightGCN [24] iteratively propagates the user embedding $(e_l)_u$ and item embedding $(e_l)_i$ as follows:

$$(e_{l+1})_u = \frac{1}{\sqrt{|\mathcal{N}(u)|}} \sum_{i \in \mathcal{N}(u)} \frac{1}{\sqrt{|\mathcal{N}(i)|}} (e_l)_i, \tag{3}$$

$$(e_{l+1})_i = \frac{1}{\sqrt{|\mathcal{N}(i)|}} \sum_{i \in \mathcal{N}(i)} \frac{1}{\sqrt{|\mathcal{N}(u)|}} (e_l)_u. \tag{4}$$

The embedding propagation of LightGCN can be re-written in matrix form as follows:

$$E_{l+1} = \tilde{A} E_l, \quad \forall l = 0, \ldots, L-1 \tag{5}$$

$$Y = \frac{1}{L+1} \sum_{l=0}^{L} E_l, \tag{6}$$

where $L$ denotes the number of GNN layers, and $Y$ denotes the model output of LightGCN with layer-wise combination. As LightGCN computes full embedding propagation in Eq. (5) for $L$ times to capture $L$-hop neighborhood information, the computation complexity of LightGCN in one training iteration is $O(L|\mathcal{E}|d)$ with the support of sparse matrix multiplications. Thus, the computation complexity for one training epoch is $O(\frac{1}{B}L|\mathcal{E}|^2 d)$.

**PinSAGE**. The embedding propagation in LightGCN aggregates all the neighbors for a user or an item, which is less compatible with Web-scale item-to-item recommender systems. Another important embedding propagation rule in GNN-based recommendation is proposed in PinSAGE:

$$(\boldsymbol{n}_{l+1})_u = \text{Aggregate}(\{\text{ReLU}(\boldsymbol{Q} \cdot (\boldsymbol{e}_l)_v + \boldsymbol{q}) \mid v \in \hat{\mathcal{N}}(u)\}), \quad (7)$$

$$(\boldsymbol{e}_{l+1})_u = \text{Normalize}(\boldsymbol{W} \cdot \text{Concat}[(\boldsymbol{e}_l)_u; (\boldsymbol{n}_{l+1})_u] + \boldsymbol{w}), \quad (8)$$

where $\boldsymbol{Q}, \boldsymbol{q}, \boldsymbol{W}, \boldsymbol{w}$ are trainable parameters, and $\hat{\mathcal{N}}(u)$ denotes the randomly sampled neighbors for node $u$. If PinSAGE constantly samples $D$ random neighbors for each node at each layer, and the sampled $B$ edges have $n_B$ target nodes without repetition, the computation complexity in each training iteration is $O(n_B D^L d^2)$ as discussed in previous studies [52]. Thus, the time complexity in the entire training epoch is $O(\frac{1}{B} n_B |\mathcal{E}| D^L d^2) = O(|\mathcal{E}| D^L d^2)$. Moreover, the neighbor sampling in PinSAGE incurs large approximation error that impacts the prediction performance.

**Matrix Factorization (MF)**. Matrix factorization and its neural variant NCF [26] are simple but strong baselines for recommendations at scale. Given learnable user embedding $\boldsymbol{p}_u$ and item embedding $\boldsymbol{q}_i$, MF models their interaction directly by inner product as $\hat{y}_{u,i} = \boldsymbol{p}_u^T \boldsymbol{q}_i$, while NCF models the interaction by deep neural networks as follows:

$$\boldsymbol{e}_L = \boldsymbol{W}_L(\cdots \phi(\boldsymbol{W}_2 \phi(\boldsymbol{W}_1 \begin{bmatrix} \boldsymbol{p}_u \\ \boldsymbol{q}_i \end{bmatrix} + \boldsymbol{b}_1) + \boldsymbol{b}_2) \cdots) + \boldsymbol{b}_L, \quad (9)$$

$$\hat{y}_{u,i} = \sigma(\boldsymbol{h}^T \boldsymbol{e}_l), \quad (10)$$

where $\boldsymbol{W}_l, \boldsymbol{b}_l$ and $\boldsymbol{h}_l$ are trainable parameters, and $\phi$ is a non-linear activation function. In each training iteration, the computation complexity for MF and NCF is $O(Bd)$ and $O(BLd^2)$, which stands for the complexity of dot products and MLPs, respectively. Thus, the time complexity in each training epoch for MF and NCF is $O(|\mathcal{E}|d)$ and $O(|\mathcal{E}|Ld^2)$.

**Inefficiency of GNNs.** In comparison with conventional MF models, GNNs' inefficiency lies behind their non-linear complexity with respect to the number of edges $|\mathcal{E}|$ or layers $L$. For example, the time complexity for LightGCN is $O(\frac{1}{B}L|\mathcal{E}|^2 d)$, which grows quadratically with $|\mathcal{E}|$, and PinSAGE has a complexity of $O(|\mathcal{E}|D^L d^2)$, which grows exponentially with $L$. In this paper, we pursue a linear-time design for GNNs, which means the time complexity of our proposed model is expected to be $O(C|\mathcal{E}|d)$, where $C$ is a small constant (e.g., $C = Ld$ for NCF).

## 3 THE PROPOSED METHOD

In web-scale recommender systems, the problem size can easily reach a billion scale towards the numbers of nodes and edges in the interaction graphs [30, 31, 45]. Therefore, it is essential that scalable recommender systems should have nearly linear or sub-linear complexity with respect to the problem size. Otherwise, they are infeasible in practice due to the unaffordable computational cost [44]. The scalability issue of GNN-based recommendation models inspires us to pursue a more efficient algorithm design with linear computation complexities. However, it is highly non-trivial to significantly reduce the computation complexity while preserving the long-range modeling ability of GNNs. First, it is

widely acknowledged that GNNs' capability of capturing long-range dependencies lies behind stacking embedding propagation layers, but the iterative propagation process makes achieving linear time complexity challenging. Second, many GNN-based recommendation models face high computational costs, and some exhibit quadratic dependency on the number of edges, as discussed in Section 2.3. In light of these challenges, neighbor sampling becomes an indispensable technique in order to achieve linear complexity. However, it's important to be aware that sampling will naturally introduce approximation errors which can impact performance. Many existing variance reduction techniques, such as [5, 8], tend to lower variance but often at the expense of increased computational costs, which may not be practical in real-world recommendation.

In Section 3.1, we tackle the challenge of balancing strong expressive power with the need for multiple aggregation layers in recommendation systems. To address this, we propose the use of implicit graph modeling to capture high-order user-item interactions efficiently. Our approach leverages historical computations, requiring just a single propagation layer to achieve this goal. In Section 3.2, we introduce a efficient approach for variance-reduced neighbor sampling, characterized by linear complexity. This approach offers a substantial reduction in computational overhead compared to existing methods while still achieving effective variance reduction.

### 3.1 Implicit Modeling for Recommendations

Personalized PageRank [36] is a classic approach for the measurement of the proximity between nodes in a graph. It is adopted by a popular GNN model, PPNP (Personalized Propagation of Neural Predictions) [20], to propagate node embeddings according to the personalized PageRank matrix:

$$\mathbf{E}_{PPNP}^k = \alpha \left( \mathbf{I} - (1 - \alpha)\tilde{\mathbf{A}} \right)^{-1} \mathbf{E}_{in}^k, \quad (11)$$

where $\alpha$ is the teleport factor and $\mathbf{E}_{in}^k$ is the input node embedding. Due to the infeasible cost of matrix inversion, APPNP approximates this by $L$ propagation layers:

$$\mathbf{E}_0^k = \mathbf{E}_{in}^k \quad (12)$$

$$\mathbf{E}_{l+1}^k = (1 - \alpha)\tilde{\mathbf{A}}\mathbf{E}_l^k + \alpha \mathbf{E}_{in}^k, \ \forall l = 0, \dots, L - 1 \quad (13)$$

such that it can capture the $L$-hop high-order information in the graph without suffering from over-smoothing due to the teleport term $\alpha \mathbf{E}_{in}^k$. LightGCN exhibits a close relation with APPNP although the embedding from different propagation layers is averaged with a different weight (see the analysis in Section 3.2 in [24]). However, like most GNNs, both APPNP and LightGCN suffer from scalability issues due to the multi-layer recursive feature aggregations, which greatly limit their applications in large-scale recommendations.

Motivated by the implicit modeling in Neural ODE [7], Deep Equilibrium Model [1], Implicit Deep Learning [11], and Implicit GNNs [21, 37, 54], we propose an implicit modeling for graph-based recommendations:

$$\mathbf{E}_{in}^k = \frac{1}{\alpha} \left( \mathbf{I} - (1 - \alpha)\tilde{\mathbf{A}} \right) \mathbf{E}_{out}^k, \quad (14)$$

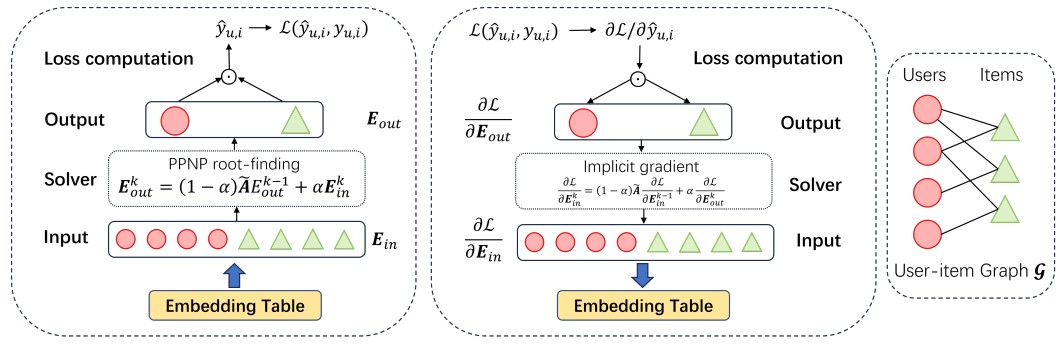

**Figure 1: An illustration of our model architecture. The forward process of our model aims to solve the PPNP fixed-point equation, which expresses an equilibrium state of the embedding propagations, and can be used to capture long-range relations between any pair of nodes regardless of their distance. The backward process computes the implicit gradients w.r.t the PPNP fixed-point and updates the input embeddings. With appropriate initialization, both forward and backward can be computed with only one propagation step.**

where the relation between output embedding $\mathbf{E}_{out}^k$ and input embedding $\mathbf{E}_{in}^k$ is implicitly defined by this fix-point equation. In other words, there is no explicit formula for the output embedding $\mathbf{E}_{out}^k$. This implicit modeling provides flexibility for the fix-point solver since the training and inference of implicit deep learning models are agnostic to the computation trajectory. In other words, we can use any root-find solver to construct the propagation layers without worrying about the compatibility with back-propagation. Specifically, to pave the way to linear-time computation, we propose to solve this fix-point equation by a single forward propagation layer:

$$\mathbf{E}_{out}^k = (1-\alpha)\tilde{\mathbf{A}}\mathbf{E}_{out}^{k-1} + \alpha\mathbf{E}_{in}^k \tag{15}$$

where $\mathbf{E}_{out}^{k-1}$ is the output embedding from iteration $k-1$ and serves as a better initialization for the fix-point solver. This single-layer design significantly reduces the computation cost of multi-layer propagation but still captures multi-hop neighbor information through information accumulation.

The backward propagation of implicit deep learning models is independent of the forward computation [1, 11, 21, 54]. Given the gradient from the output embedding layer $\frac{\partial \mathcal{L}}{\partial \mathbf{E}_{out}^k}$, the gradient of $\mathbf{E}_{in}^k$ can be computed based on the fix-point equation in Eq. (14):

$$\frac{\partial \mathcal{L}}{\partial \mathbf{E}_{in}} = \alpha\frac{\partial \mathcal{L}}{\partial \mathbf{E}_{out}}\left(\mathbf{I} - (1-\alpha)\tilde{\mathbf{A}}\right)^{-1}. \tag{16}$$

Due to the prohibitively high dimensionality of the adjacency matrix, computing its inverse is infeasible. Therefore, we propose to approximate this gradient by a single backward propagation layer:

$$\frac{\partial \mathcal{L}}{\partial \mathbf{E}_{in}^k} = (1-\alpha)\tilde{\mathbf{A}}\frac{\partial \mathcal{L}}{\partial \mathbf{E}_{in}^{k-1}} + \alpha\frac{\partial \mathcal{L}}{\partial \mathbf{E}_{out}^k} \tag{17}$$

where $\frac{\partial \mathcal{L}}{\partial \mathbf{E}_{in}^{k-1}}$ is the gradient of input embedding from iteration $k-1$ and serves as a better initialization. In summary, the forward and backward computation of our single-layer GNN are formulated as:

$$\textbf{Forward}: \mathbf{E}_{out}^k = (1-\alpha)\tilde{\mathbf{A}}\mathbf{E}_{out}^{k-1} + \alpha\mathbf{E}_{in}^k \tag{18}$$

$$\textbf{Backward}: \frac{\partial \mathcal{L}}{\partial \mathbf{E}_{in}^k} = (1-\alpha)\tilde{\mathbf{A}}\frac{\partial \mathcal{L}}{\partial \mathbf{E}_{in}^{k-1}} + \alpha\frac{\partial \mathcal{L}}{\partial \mathbf{E}_{out}^k} \tag{19}$$

where $\mathbf{E}_{out}$ and $\frac{\partial \mathcal{L}}{\partial \mathbf{E}_{in}}$ are two auxiliary variables to be maintained in the memory.

## 3.2 Efficient Variance-Reduced Neighbor Sampling

The implicit modeling and single-layer design introduced in Section 3.1 significantly reduce the computation complexity (per training epoch) of LightGCN from $O(\frac{L|\mathcal{E}|^2 d}{B})$ to $O(\frac{|\mathcal{E}|^2 d}{B})$. However, it does not reach the linear complexity as Matrix Factorization methods due to the quadratic dependency on the number of edges $|\mathcal{E}|$. This quadratic dependency is caused by the full neighbor aggregation. Therefore, the key solution to tackle this quadratic dependency is neighbor sampling as initially proposed in GraphSAGE [23] and PinSAGE [56]. Practically, neighbor sampling is urgently needed for real-world large-scale recommendation systems where the degree distribution of user and item nodes often follows a power-law distribution [16, 55]. This implies that trending items and active users could have thousands of interaction records, which incurs significant computation costs for their embedding loading and aggregation.

Unfortunately, neighbor sampling will cause large approximation errors and suffer from performance degradation as demonstrated in large-scale OGB benchmarks [10]. Exemplified by VR-GCN [5] and MVS-GNN [8], variance-reduction (VR) techniques have been introduced to reduce the approximation error in graph convolutions. However, we will reveal that these methods still require the full embedding aggregation of historical embedding, which maintains the undesirable quadratic dependency on $|\mathcal{E}|$. To this end, we will propose an efficient variance-reduced neighbor sampling approach to achieve linear complexity.

**Classic Variance-reduced Neighbor Aggregation.** Recent research has investigated variance reduction on GNNs, such as VR-GCN and MVS-GNN [5, 8]:

$$(E_{out}^k)^{VR} = \hat{A}[(E_{in}^k) - (\overline{E}_{in}^k)] + \tilde{A}(\overline{E}_{in}^k) \tag{20}$$

where $\hat{A}$ is an unbiased estimator of $\tilde{A}$, $\hat{A}_{u,v} = \frac{|\mathcal{N}(u)|}{D}\tilde{A}_{u,v}$ if node $v$ is sampled as a neighbor of node $u$, otherwise $\hat{A}_{u,v} = 0$. $\overline{E}_{in}^k$ is the historical embeddings for approximating $E_{in}^k$. However,

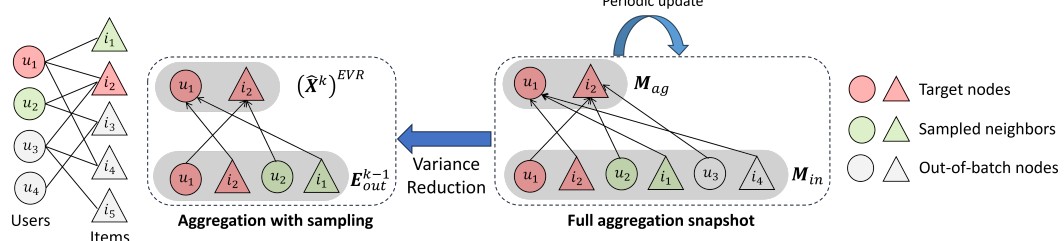

**Figure 2: The Process of Efficient Variance-Reduced Neighbor Sampling in LTGNN.**

such approaches need to perform full neighbor aggregations on the historical embedding by computing $\tilde{A}(\bar{E}_{in}^{k})$. Importantly, this computation has to be performed in each mini-batch iteration, leading to the quadratic computation complexity $O(\frac{|\mathcal{E}|^2 d}{B})$ for the whole training epoch. Therefore, they seriously sacrifice the computational efficiency of neighbor sampling in large-scale recommender systems.

**Efficient Variance-reduced Neighbor Sampling.** To further reduce the quadratic computation complexity, we propose to compute the historical embedding aggregation periodically instead of computing them in every training iteration. Specifically, we allocate two memory variables $M_{in}$ and $M_{ag}$ to store the historical input embedding and fully aggregated embedding, where $M_{ag} = \tilde{A}M_{in}$. The input memory variable $M_{in}$ is updated periodically at the end of each training epoch, and the aggregated embedding $M_{ag}$ are updated based on the renewed inputs. We name it as Efficient Variance Reduction (EVR), which can be formulated as:

$$(\hat{X}^k)^{EVR} = \hat{A}[(E_{out}^{k-1}) - M_{in}] + M_{ag} \tag{21}$$

$$(\hat{E}_{out}^k)^{EVR} = (1 - \alpha)(\hat{X}^k)^{EVR} + \alpha(E_{in}^k). \tag{22}$$

The whole process of our algorithm is shown in Figure 2.

**Complexity analysis.** If we denote the number of sampled neighbors for each node as $D$ and assume the sampled $B$ edges have $n_B$ target nodes without repetition, then the complexity of our methods is $O(\frac{|\mathcal{E}|n_B D d}{B}) = O(|\mathcal{E}|Dd)$ since $n_B$ and $B$ are of the same order. Given that $D$ is a constant, the complexity no longer depends on the number of edges in a quadratic manner, and instead, it becomes linear. As a result, it significantly reduces the computational cost associated with our methods. With our variance-reduced neighbor sampling techniques, we can avoid the costly full aggregation computation at each training iteration by a periodic update with linear computational complexity. This reduction in complexity is highly significant, as shown in the complexity comparison in Table 1.

**Table 1: Complexity Comparisons.**

| Models | Computation Complexity |
|---|---|
| LightGCN | $O(\frac{1}{B}L|\mathcal{E}|^2 d)$ |
| PinSAGE | $O(|\mathcal{E}|D^L d^2)$ |
| MF | $O(|\mathcal{E}|d)$ |
| NCF | $O(L|\mathcal{E}|d^2)$ |
| LTGNN | $O(|\mathcal{E}|Dd)$ |

REMARK 1. *Prior works on variance reduction [3, 5, 37] mainly concentrate on reducing the variance of the forward process. In*

*contrast, our variance reduction method not only significantly reduces the computation cost but also is symmetrically applied to both forward and backward processes in embedding sampling and gradient sampling similar to Eq. (21) and Eq. (22). This achieves variance reduction in both forward and backward propagations.*

## 4 EXPERIMENTS

In this section, we will verify the effectiveness and efficiency of the proposed LTGNN framework with comprehensive experiments. Specifically, we aim to answer the following research questions:

- **RQ1**: Can LTGNN achieve promising prediction performance on large-scale recommendation datasets? (Section 4.2)
- **RQ2**: Can LGTNN handle large user-item interaction graphs more efficiently than existing GNN approaches? (Section 4.3)
- **RQ3**: How does the effectiveness of the proposed LTGNN vary when we ablate different parts of the design? (Section 4.4)

### 4.1 Experimental Settings

We first introduce the datasets, baselines, evaluation metrics, and hyperparameter settings as follows.

**Datasets.** We evaluate the proposed LTGNN and baselines on two medium-scale datasets including *Yelp2018* and *Alibaba-iFashion*, and one large-scale dataset *Amazon-Large*. Yelp2018 dataset is released by the baselines NGCF [48] and LightGCN [24], and the Alibaba-iFashion dataset can be found in the GitHub repository[1]. For the large-scale setting, we construct the large-scale dataset, Amazon-Large, based on the rating files from the Amazon Review Data website[2]. Specifically, we select the three largest subsets (i.e., Books, Clothing Shoes and Jewelry, Home and Kitchen) from the entire Amazon dataset, and then keep the interactions from users who are shared by all the three subsets (7.9% of all the users). The rating scale is from 1 to 5, and we transform the explicit ratings into implicit interactions by only keeping the interactions with ratings bigger than 4. To ensure the quality of our Amazon-Large dataset, we follow a widely used 10-core setting [27, 46, 48] and remove the users and items with interactions less than 10 by following the data preparation settings [26, 27, 47]. The statistical summary of the datasets can be found in Table 2.

**Table 2: Dataset statistics.**

| Dataset | # Users | # Items | # Interactions |
|---|---|---|---|
| Yelp2018 | 31, 668 | 38, 048 | 1, 561, 406 |
| Alibaba-iFashion | 300, 000 | 81, 614 | 1, 607, 813 |
| Amazon-Large | 872, 557 | 453, 553 | 15, 236, 325 |

[1]https://github.com/wenyuer/POG
[2]https://cseweb.ucsd.edu/~jmcauley/datasets/amazon_v2/

**Baselines.** The main focus of this paper is to enhance the scalability of GNN-based collaborative filtering methods. Therefore, we compare our method with the most widely used GNN backbone in recommendations, *LightGCN* [24] and its scalable variants that employ typical GNN scalability techniques, including GraphSAGE [23], VR-GCN [5] and GAS [18]. The corresponding variants of LightGCN are denoted as *LightGCN-NS* (for Neighbor Sampling), *LightGCN-VR*, and *LightGCN-GAS*.

To demonstrate the effectiveness of our method, we also compare it with a range of representative recommendation models, including MF [35], NCF [26], GC-MC [2], PinSAGE [56], NGCF [48], and DGCF [24]. In this paper, we adopt two widely used evaluation metrics in recommendations: Recall@K and Normalized Discounted Cumulative Gain (NDCG@K) [24, 48]. We set the K=20 by default, and we report the average result for all test users. Moreover, since we are designing an efficient GNN-based collaborative filtering backbone that is independent of the loss function, our method is orthogonal to SSL-based methods [39, 51, 57] and negative sampling algorithms [29, 40]. We will explore the combination of our method and these orthogonal designs in future work.

**Parameter Settings.** We implement the proposed LTGNN using PyTorch and PyG libraries. We strictly follow the settings of NGCF [48] and LightGCN [24] to implement our method and the scalable LightGCN variants for a fair comparison. All the methods use an embedding size of 64, a BPR batch size of 2048, and a negative sample size of 1. For the proposed LTGNN, we tune the learning rate from {5e-4, 1e-3, 1.5e-3, 2e-3} and the weight decay from {1e-4, 2e-4, 3e-4}. We employ an Adam [33] optimizer to minimize the objective function. For the coefficient $\alpha$ in PPNP, we perform a grid search over the hyperparameter in the range of [0.3, 0.7] with a step size of 0.05. To ensure the scalability of our model, the number of propagation layers $L$ is fixed to 1 by default, and the number of sampled neighbors $d$ is searched in {5, 10, 15, 20}. For the GNN-based baselines, we follow their official implementations and suggested settings in their papers. For the LightGCN variants with scalability techniques, the number of layers $L$ is set based on the best choice of LightGCN, and we search other hyperparameters in the same range as LTGNN and report the best results.

## 4.2 Recommendation Performance

In this section, we mainly examine the recommendation performance of our proposed LTGNN, with a particular focus on comparing LTGNN with the most widely adopted GNN backbone LightGCN. We use out-of-memory (OOM) to indicate the methods that cannot run on the dataset due to memory limitations. The recommendation performance summarized in Table 3 provides the following observations:

- Our proposed LTGNN achieves comparable or better results on all three datasets compared to the strongest baselines. In particular, LTGNN outperforms all the baselines on Yelp and Alibaba-iFashion. The only exception is that the Recall@20 of LightGCN (L=3) outperforms LTGNN (L=1) on the Amazon-Large dataset. However, our NDCG@20 outperforms LightGCN (L=3), and LTGNN (L=1) is much more efficient compared with LightGCN (L=3), as LTGNN

only uses one embedding propagation layer and very few randomly sampled neighbors.

- The scalable variants of LightGCN improve the scalability of LightGCN by sacrificing its recommendation performance in most cases. For instance, the results for LightGCN-VR, LightGCN-NS, and LightGCN-GAS are much worse than LightGCN with full embedding propagation on Amazon-Large. In contrast, the proposed LTGNN has better efficiency than these variants and preserves the recommendation performance.

- The performance of GNN-based methods like NGCF and LightGCN consistently outperforms earlier methods like MF. However, GNNs without scalability techniques can hardly be run large-scale datasets. For instance, GC-MC, NGCF, and DGCF significantly outperform MF, but they are reported as OOM on the Amazon-Large dataset. This suggests the necessity of pursuing scalable GNNs for improving the recommendation performance in large-scale industry scenarios.

## 4.3 Efficiency Analysis

To verify the scalability of LTGNN, we provide efficiency analysis in comparison with MF, LightGCN, and scalable variants of LightGCN with different layers on on two large-scale datasets: Alibaba-iFasion and Amazon-Large. From the running time shown in Table 4, we can make the following conclusions:

- Our proposed single-layer LTGNN has a comparable running time compared with one-layer LightGCN with sampling, and it is faster than the original LightGCN. This aligns with our complexity analysis presented in Section 3.2. Furthermore, LTGNN is faster than one-layer LightGCN with variance reduction, thanks to our improved and efficient variance reduction (EVR) techniques. Note that the accuracy of one-layer LightGCN is much worse than LTGNN as shown in Figure 3.

- LTGNN demonstrates significantly improved computational efficiency compared to baseline models with more than one layer. When combined with the results from Table 3, it becomes evident that LTGNN can maintain high performance while achieving a substantial enhancement in computational efficiency.

- While the running time of LTGNN is a few times longer than that of Matrix Factorization (MF) due to their constant difference in the complexity analysis, it's important to note that LTGNN already achieves a nice and *similar scaling behavior as MF* without undesirable dependency on the number of edges or layers. Therefore, it scales much better than other GNN-based methods. Our observations are consistent with the complexities in Table 3.

- An interesting observation is that on large-scale datasets, full-graph LightGCN outperforms LightGCN with neighbor sampling on efficiency. This is mainly because of the high CPU cost of random sampling, which restraints the usage rate of GPUs.

**Table 3: The comparison of overall prediction performance.**

| Dataset | Yelp2018 | | Alibaba-iFashion | | Amazon-Large | |
|---|---|---|---|---|---|---|
| Method | Recall@20 | NDCG@20 | Recall@20 | NDCG@20 | Recall@20 | NDCG@20 |
| MF | 0.0436 | 0.0353 | 0.05784 | 0.02676 | 0.02752 | 0.01534 |
| NCF | 0.045 | 0.03640 | - | - | - | - |
| GC-MC | 0.0462 | 0.0379 | 0.07738 | 0.03395 | OOM | OOM |
| PinSAGE | 0.0495 | 0.04049 | - | - | - | - |
| NGCF | 0.0581 | 0.0475 | 0.07979 | 0.0357 | OOM | OOM |
| DGCF | 0.064 | 0.0522 | 0.08445 | 0.03967 | OOM | OOM |
| LightGCN (L=3) | 0.06347 | 0.05238 | 0.08793 | 0.04096 | 0.0331 | 0.02283 |
| LightGCN-NS (L=3) | 0.06256 | 0.0514 | 0.08804 | 0.04147 | 0.02835 | 0.02035 |
| LightGCN-VR (L=3) | 0.06245 | 0.05141 | 0.08814 | 0.04082 | 0.02903 | 0.02093 |
| LightGCN-GAS (L=3) | 0.06337 | 0.05207 | 0.08169 | 0.03869 | 0.02886 | 0.02085 |
| LTGNN (L=1) | 0.06393 | 0.05245 | 0.09335 | 0.04387 | 0.02942 | 0.02585 |

**Table 4: The comparison of running time (s) on three datasets.**

| Dataset | | Alibaba-iFashion | Amazon-Large |
|---|---|---|---|
| Method | # Layer | Runnning Time | Running Time |
| LightGCN | | 51.4s | 2999.35s |
| LightGCN-NS | $L = 3$ | 51.70s | 4291.37s |
| LightGCN-VR | | 59.79s | 4849.59s |
| LightGCN-GAS | | 26.576s | 932.03s |
| LightGCN | | 30.78s | 2061.75s |
| LightGCN-NS | $L = 2$ | 26.89s | 1305.25s |
| LightGCN-VR | | 30.33s | 1545.34s |
| LightGCN-GAS | | 25.04s | 903.78s |
| LightGCN | | 18.02s | 1117.51s |
| LightGCN-NS | $L = 1$ | 12.74s | 684.84s |
| LightGCN-VR | | 13.92s | 870.82s |
| LightGCN-GAS | | 13.35s | 729.22s |
| MF | - | 4.60s | 127.24s |
| LTGNN | $L = 1$ | 13.68s | 705.91s |

## 4.4 Ablation Study

In this section, we provide extensive ablation studies to evaluate the effectiveness of different parts in our proposed framework.

**Effectivenss of implicit graph modeling.** We conduct an ablation study to show the effect of embedding propagation layers and long-range collaborative signals. Particularly, we use the same setting for LightGCN and LTGNN and change the number of propagation layers $L$. As illustrated in Figure 3, we have two key observations: 1) LTGNN can reach better performance in comparison with LightGCN with only one or two propagation layers, which demonstrates the strong long-range modeling capability of our proposed model; 2) Adding more propagation layers into LTGNN will not significantly improve its performance, which means $L = 1$ is the best choice for LTGNN to balance its performance and scalability.

**Effectiveness of efficient variance reduction.** In this study, we aim to demonstrate the effectiveness of our proposed EVR algorithm by showing the impact of the number of neighbors on recommendation performance. As shown in Figure 4, LTGNN with efficient variance reduction consistently outperforms its vanilla neighbor sampling variant (i.e., LTGNN-NS) regardless of the number of neighbors, illustrating its effect in reducing the large approximation error caused by neighbor sampling. The recommendation performance of LTGNN with efficient variance reduction is remarkably stable, even under extreme conditions like sampling only 2 neighbors for each target node. This indicates the great potential of our proposed LTGNN in large-scale recommendations, as a GNN with only one propagation layer and two random neighbors will be ultra-efficient compared with previous designs that incur a large number of layers and neighbors.

**Numerical Analysis.** In this experiment, we compute the PPNP embedding propagation result $E_{PPNP}^k$ for the target nodes as an indicator of long-range modeling ability as it serves as fixed-point, and we compute the relative error between the model output $E_L^k$ and the this PPNP computation result. We use $L = 1$ for LTGNN and its two variants - LTGNN-NS and LTGNN-Full, which denotes LTGNN without efficient variance reduction and LTGNN with exact full neighbor aggregation. From Figure 5, we have two observations as follows: 1) On both datasets, the output of LTGNN converges to PPNP after 4000 training iterations (i.e., less than training 10 epochs), which means our proposed LTGNN can capture the long-range dependencies in user-item graphs by using only one propagation layer; 2) By comparing LTGNN with its variants, it is obvious that neighbor sampling without variance reduction seriously hurts the modeling ability of LTGNN, and our proposed LTGNN has similar convergence curves in comparison to LTGNN with full aggregation, showing the effectiveness of our proposed efficient variance reduction method.

## 5 RELATED WORK

In this section, we summarize the related works on graph-based collaborative filtering and scalable GNNs.

### 5.1 Graph Collaborative Filtering Models for Recommendations

In modern recommender systems, collaborative filtering (CF) is one of the most represeative paradigm [4, 26] to understand users preferences. The basic idea of CF is that users with similar historical behaviors are more likely to share similar preferences toward items [43]. Among various CF techniques, MF is proposed

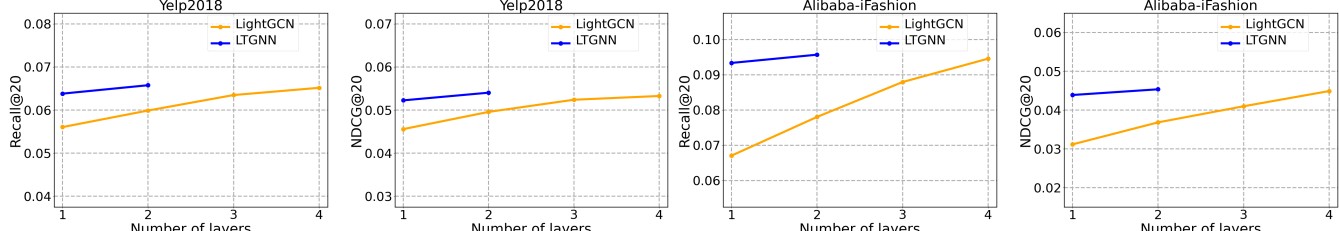

**Figure 3: Performance comparison between LTGNN and LightGCN using different layers on Yelp2018 and Alibaba-iFashion.**

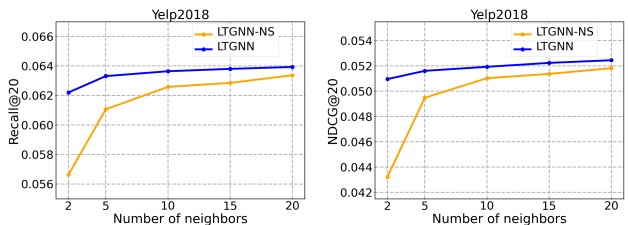

**Figure 4: Performance of a 1-layer LTGNN w.r.t different numbers of sampled neighbors on Yelp2018.**

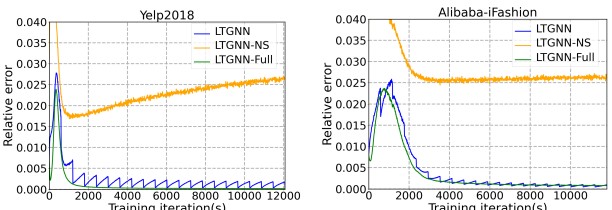

**Figure 5: The relative error between the model output $E_L^k$ and the exact PPNP propagation result $E_*^k$ of the embeddings (i.e., $||E_L^k - E_*^k||_F / ||E_*^k||_F$).**

to decompose the user-item interaction data into trainable embeddings for users and items, and then reconstruct the missing interactions [15, 26, 35]. Early works in CF mainly model the user-item interactions with scaled dot-product [4, 35], MLPs [13, 26], and LSTMs [15, 22]. However, these models fail to model the high-order collaborative signals between users and items, leading to sub-optimal representations of users and items.

In recent years, a promising line of studies has incorporated GNNs into CF-based recommender systems. The key advantage of utilizing GNN-based recommendation models is that GNNs can easily capture long-range dependencies via the information propagation mechanism on the user-item graph. For example, an early exploration, GC-MC, completes the missing user-item interactions with graph convolutional autoencoders [2]. For large-scale recommendation scenarios, PinSAGE [56] adapts the key idea of GraphSAGE [23] to recommendations and achieves promising results. Another significant milestone in GNN-based recommendations is the NGCF [48], which explicitly formulates the concept of collaborative signals and models them as high-order connectivities by message-passing propagations. Afterward, LightGCN [24] indicates the non-linear activation functions and feature transformations in GNN-based recommender systems are

redundant, and proposes to simplify existing GNNs while achieving promising performance. However, despite the success of previous GNN-based models in capturing user preferences, existing works fail to address the neighborhood explosion problem on large-scale recommendation scenarios, which indicates that the scalability of GNN-based recommender systems remains an open question.

## 5.2 Scalability of Graph Neural Networks

Recently, extensive literature has studied the efficiency and scalability of GNNs on large-scale graphs. Various novel paradigms are introduced to improve the scalability of GNNs, including sampling methods, pre-computing methods, post-computing methods, and memory-based methods. Sampling-based methods lower computation and memory requirements using mini-batch training strategies on GNNs. The well-known neighborhood explosion problem can be addressed by only keeping a limited number of neighbors [3, 5, 23] or updating by feature memory[18, 54]. Pre-computing or post-computing methods decompose the end-to-end training process of GNNs into two stages: embedding propagation and prediction. In particular, pre-computing methods pre-calculate the embedding aggregation results before training the prediction model [19, 50, 58], while post-computing methods firstly train a feature transformation model and leverage unsupervised feature propagation methods for prediction [28, 59]. In retrospect, sampling methods and pre-computing/post-computing methods may still encounter high computational costs, introduce substantial approximation errors, or sacrifice the advantage of end-to-end training. These limitations can significantly constrain the potential of GNNs in handling billion-scale of users and items on real-world recommender systems.

## 6 CONCLUSION

Scalability is a major challenge for GNN-based recommender systems, as they often require high computational resources to handle large-scale recommendation scenarios. To address this challenge, we propose a novel scalable GNN model for recommendation, which leverages implicit graph modeling and variance-reduced neighbor sampling to capture long-range collaborative signals. The proposed LTGNN only needs one propagation layer and a fixed number of one-hop neighbors, which reduces the computation complexity to be linear to the number of edges, showing great potential in industrial recommendation applications. Extensive experiments on three real-world datasets are conducted to demonstrate the effectiveness and efficiency of our proposed model. We believe it will significantly broaden the impact of GNN-based methods for real-world recommendation systems.

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
