# OpenReview forum: "Linear-Time Graph Neural Networks for Scalable Recommendations"
_ACM.org/TheWebConf/2024/Conference — TheWebConf24_

### Official Review · Reviewer_W5Tr · 2023-11-18

**Novelty:** 4
**Technical Quality:** 5

**Review:**

**Summary:**

The authors propose a Linear-Time GNN (LTGNN) that can handle large-scale user-item interaction graphs with linear computation complexity and strong expressive power. LTGNN uses implicit graph modeling and efficient variance-reduced neighbor sampling to achieve this goal.

**Strengths:**

1. The authors introduce some notations and definitions of user-item interaction graphs, mini-batch training, and embedding propagation.
2. They also present the details of LTGNN, including the implicit modeling for capturing high-order information with a single propagation layer, and the efficient variance-reduced neighbor sampling for reducing the neighbor size and the approximation error. They also provide complexity analysis and comparison to show the superiority of LTGNN over existing methods.
3. They also conduct extensive experiments and ablation studies on three real-world recommendation datasets, including a large-scale dataset with millions of users and items. The results demonstrate that LTGNN can significantly reduce the training time of GNN-based recommendation models while preserving or improving the recommendation performance. They also answer some research questions and provide insights into the effectiveness and scalability of LTGNN.

**Weaknesses:**

1. The presentation of this paper requires significant improvement. There are numerous grammatical errors and typographical mistakes present throughout. For example, "a efficient"->"an efficient"; " into learning user and item representations"->" into the user and item representations".
2. The experimental baseline selected in this article is too old, and the latest Baseline DGCF was proposed in 2020. It means that the author did not follow the latest research progress, which is unacceptable in the rapidly changing field of recommender systems.
3. In Table 3, the author not only defaults a large number of values but also directly puts "OOM" in the results in part of the baseline. I think this is an imprecise experiment.

**Questions:**

1. The authors claim that his algorithm makes "the complexity no longer depends on the number of edges in a quadratic manner", but as we all know, graph networks have sparse graphs and dense graphs. When the graph becomes sparse, the number of edges will tend to be closer to the nodes. Number, will LTGNN degrade? Could the authors provide experiments on sparse graphs?
2. Referring to the series of "Weaknesses" I mentioned, could the authors give any solutions?

**Reviewer Confidence:**

3: The reviewer is confident but not certain that the evaluation is correct

**Scope:**

3: The work is somewhat relevant to the Web and to the track, and is of narrow interest to a sub-community

---

### Official Review · Reviewer_vMYX · 2023-11-23

**Novelty:** 6
**Technical Quality:** 5

**Review:**

# Summary

The authors present an approach based on Graph Neural Networks (GNNs) that is called Linear Time GNN (LTGNN) to improve the computational complexity for training GNNs as recommender models while keeping model expressiveness high to outperform more classical methods. To validate their claims, they show various offline evaluation experiments on industrial datasets and ablation studies to assess scalability and accuracy of their approach. The proposed solution includes two components, a single-layer propagation model design and a variance-reduced neighbor sampling algorithm.

# Strengths

- The paper clearly outlines the benefit of the approach which is to reduce computational complexity while improving or maintaining prediction performance.
- The paper successfully derives a GNN-approach towards recommendations with linear computational complexity for training.
- Compared to the highest baselines the margins of prediction outperformance are still substantial even though reporting the variance of those results could build additional trust in the superiority of the derived LTGNN-approach.


# Weaknesses
## Minor
1. There were no additional information on the data splitting strategy for training, validation, and testing.
2. Section 2.2 seems to overuse $\mathcal{O}$ once for denoting the dataset and once it is used to describe the training objective, where the training objective is actually $\mathcal{L}_{BPR}$. Training objective and dataset are not the same concepts
3. In section 3.1 there was a quick change from PPNP to APPNP which was unexplained and incomprehensible.
4. Figures reported in tables could benefit from using the same number of digits and vertical alignment so that visual comparison is easier - along with highlighting the best results in ea. subcategory of methods.

# Formal Comments
- Introduction: "that whether GNN"
- Section 3: an "efficient approach"

**Questions:**

- How and why are DNNs (only) classic first-order models as mentioned in the introduction?
- Are you going to release the code for your model and experiments to facilitate the reproducibility of your study?

**Ethics Review Description:**

.

**Reviewer Confidence:**

3: The reviewer is confident but not certain that the evaluation is correct

**Scope:**

4: The work is relevant to the Web and to the track, and is of broad interest to the community

---

### Official Review · Reviewer_53aS · 2023-11-27

**Novelty:** 5
**Technical Quality:** 5

**Review:**

This work proposes a Linear-Time Graph Neural Network (LTGNN) for scalable recommendations, addressing the key challenge of scaling GNN-based recommender systems to the efficiency of classic Matrix Factorization (MF) and Deep Neural Network (DNN) methods while maintaining high predictive performance. The authors leverage implicit graph modeling and efficient variance-reduced neighbor sampling to achieve a linear complexity in computation, which is critical for handling large-scale data.

Pros
- Innovative approach to scaling GNNs for recommendation systems, addressing a critical industry need.
- Empirical results demonstrate the model's effectiveness and efficiency on large-scale datasets.
- The quality and the clarity of the work are good. The ideas are easy to follow.

Cons
- The adaptability of the model to dynamically evolving datasets could be explored further.
- Real-world deployment scenarios and their challenges could be discussed more thoroughly.

**Questions:**

- How does LTGNN adapt to the evolving nature of user-item interaction data over time?

**Reviewer Confidence:**

3: The reviewer is confident but not certain that the evaluation is correct

**Scope:**

3: The work is somewhat relevant to the Web and to the track, and is of narrow interest to a sub-community

---

### Official Review · Reviewer_QRbd · 2023-11-27

**Novelty:** 4
**Technical Quality:** 4

**Review:**

Pros:
This paper has a clear starting point and motivation, and has a certain inspiration and promotion for the development of the research field.
The structure of the paper is reasonable, the writing is smooth and the readability is strong.

Cons:
First of all, the starting point of LTGNN is to obtain a GNN recommendation system that can be effectively scaled. So why only LTGNN model with layers 1 and 2 is conducted in Figure 3 instead of continuing to extend layers like LightGCN? Moreover, it can be clearly seen that when the number of layers is 2, the performances of LTGNN are still increasing.
In addition, as for the main experiment, the author only reported Recall@20 and NDCG@20. Does the conclusion also change with different K? In addition, the stability of the model effect also needs to be considered, it is best to train the model several times and report the average performance of the model.
Although the authors have reported results on three academic datasets, it would be better to have experimental results in real industry scenarios.

**Questions:**

Please look at the disadvantages. My main concern is whether the advantages of LTGNN can be maintained with different experimental settings (including datasets, model hyperparameters, and evaluation metrics). The authors need to provide more details to prove the validity of the model.

**Reviewer Confidence:**

2: The reviewer is willing to defend the evaluation, but it is likely that the reviewer did not understand parts of the paper

**Scope:**

3: The work is somewhat relevant to the Web and to the track, and is of narrow interest to a sub-community

---

### Decision · Program_Chairs · 2024-01-22

**Decision:**

Accept

**Comment:**

The reviewers largely agree that the work is novel and is of good technical quality. The authors provided extensive explanations and additional details in the discussion phase. Code and data are provided in an anonymized repository. Overall, the paper could be a solid contribution to the conference.